# Revealing low-loss dielectric near-field modes of hexagonal boron nitride by photoemission electron microscopy

Yaolong Li[1], Pengzuo Jiang[1], Xiaying Lyu[1], Xiaofang Li[1], Huixin Qi[1], Jinglin Tang[1], Zhaohang Xue[1], Hong Yang[1,2,3], Guowei Lu[1,2,3], Quan Sun [2] ✉, Xiaoyong Hu [1,2,3] ✉, Yunan Gao [1,2,3] ✉ & Qihuang Gong [1,2,3]

Low-loss dielectric modes are important features and functional bases of fundamental optical components in on-chip optical devices. However, dielectric near-field modes are challenging to reveal with high spatiotemporal resolution and fast direct imaging. Herein, we present a method to address this issue by applying time-resolved photoemission electron microscopy to a low-dimensional wide-bandgap semiconductor, hexagonal boron nitride (hBN). Taking a low-loss dielectric planar waveguide as a fundamental structure, static vector near-field vortices with different topological charges and the spatiotemporal evolution of waveguide modes are directly revealed. With the lowest-order vortex structure, strong nanofocusing in real space is realized, while near-vertical photoemission in momentum space and narrow spread in energy space are simultaneously observed due to the atomically flat surface of hBN and the small photoemission horizon set by the limited photon energies. Our approach provides a strategy for the realization of flat photoemission emitters.

Low-loss dielectric modes such as dielectric waveguide modes, topological edge states[1], metasurfaces[2] and whispering-gallery modes[3] are important features and functional bases of fundamental optical components for constructing on-chip optical devices. The near-field properties of dielectric modes, including static near-field mode distributions, dynamic mode coupling, switching, and evolution, are essential parameters for fundamental physics and device optimization. The physical processes of optical near-field modes generally occur on the nano-femto scales. However, revealing dielectric near-field modes with high spatiotemporal resolution and fast direct imaging remains a challenge. In general, techniques used to probe near-field modes include scanning near-field optical microscopy (SNOM), cathodoluminescence (CL), electron energy loss spectroscopy (EELS), and photoemission electron microscopy (PEEM)[4–6], etc. Among them,

time-resolved PEEM (TR-PEEM) has the advantage of achieving nano-femto spatiotemporal resolution and fast direct imaging. Additionally, PEEM can image photoemitted electrons in momentum and energy spaces. In addition, the photon excitation working process of PEEM causes less damage to samples than electron excitation with CL and EELS, and the laser parameters of the input pulses can be flexibly controlled.

TR-PEEM has been successfully applied to investigate near-field modes supported by metals, i.e., surface plasmon polaritons (SPPs) and surface plasmon resonances (SPRs)[7–17]. However, the inherent loss of metal inhibits their application in nanophotonics. Compared with high-loss metal nanostructures, low-loss dielectric modes have much wider applications. However, their investigations with TR-PEEM are still lacking despite a few attempts with high-loss materials such as

[1]State Key Laboratory for Mesoscopic Physics & Department of Physics, Collaborative Innovation Center of Quantum Matter and Frontiers Science Center for Nano-optoelectronics, Beijing Academy of Quantum Information Sciences, Peking University, 100871 Beijing, China. [2]Peking University Yangtze Delta Institute of Optoelectronics, 226010 Nantong, Jiangsu, China. [3]Collaborative Innovation Center of Extreme Optics, Shanxi University, 030006 Taiyuan, Shanxi, China. ✉e-mail: sunquan@ydioe.pku.edu.cn; xiaoyonghu@pku.edu.cn; gyn@pku.edu.cn

indium tin oxide (ITO) and narrow-gap semiconductors[18–20]. One major challenge is the conductivity of low-loss dielectric materials because many nonconductive materials introduce surface charging immediately and severely degrade the performance of PEEM imaging. The issue can be resolved by the careful selection of materials to balance bandgap and surface charging effects.

The development of low-dimensional van der Waals materials provides a large database for nanophotonics design[21]. Among them, hexagonal boron nitride (hBN) is a dielectric optical material with an atomically flat surface, a wide bandgap (~6 eV), and a high refractive index (in-plane $n > 2.1$) from ultraviolet to near-infrared[22–24], and it has been reported to construct low-loss dielectric modes such as Bragg grating and cavity modes[25–27]. Interestingly, we found that hBN flakes are compatible with PEEM measurements and can be used to investigate low-loss dielectric near-field modes.

In this study, we report a platform for revealing low-loss dielectric near-field modes with TR-PEEM using a low-dimensional wide-bandgap semiconductor hBN. Starting from low-loss dielectric planar waveguide structures, dielectric vector near-field vortex modes with a series of topological charges were constructed and observed at nanoscale spatial resolution. The spatiotemporal evolution of dielectric waveguide modes was directly revealed with high spatiotemporal resolution. The <80 nm localization and >$10^3$ photoemission enhancement, near-vertical photoemission, and 0.65 eV electron energy spread were realized with the designed near-field modes, implying that the hBN nanostructures could function as photoemission emitters. This study demonstrates that TR-PEEM can be a platform for investigating low-loss dielectric modes and is promising for the investigation of dielectric topological modes, metasurfaces, and device inspections.

## Results

### Static PEEM measurements

A typical design of hBN nanostructures and measurement methods for low-loss dielectric near-field modes are shown in Fig. 1. The planar waveguide was primarily constructed using hBN flakes and a glass substrate (Fig. 1a). The refractive index of hBN is anisotropic, and its in- and out-of-plane components are ~2.3 and ~1.9, respectively, within the wavelengths of 390–440 nm[23,24]. Because hBN is a wide-bandgap semiconductor, theoretically, it has no optical absorption in the

visible-to-ultraviolet range and is suitable for constructing low-loss nanophotonic structures. Only the fundamental TE modes were selectively and efficiently excited with a slit coupler when the thickness of hBN was <100 nm for an excitation wavelength of approximately 410 nm, whereas the excitation of the TM mode was negligible, as indicated by simulations and verified by PEEM experiments (Supplementary Figs. 3–6). In this study, the thickness of the hBN was selected to be between 60 and 80 nm. Therefore, the fundamental TE modes were used in the following investigations. A very thin (~10 nm) ITO layer was coated on top of the glass substrate to avoid surface charging during PEEM experiments while maintaining the low-loss characteristic of dielectric modes (discussions on the choice of ITO and charging effect are presented in Supplementary Note 5). The hBN flakes were mechanically exfoliated and transferred to the substrate, and the nanostructures on the hBN were fabricated using a focused ion beam (FIB) to produce the designed near-field modes. A typical optical image of the nanostructures with an hBN thickness of ~60 nm is shown in Fig. 1c (see Supplementary Fig. 1 for SEM image).

By using the PEEM equipped with a femtosecond laser beam at normal incidence, a pump-probe beamline at oblique incidence, and hemispherical electron energy analyzer, measurements with high spatiotemporal and 150 meV energy resolutions were achieved. Schematics of the PEEM measurements are shown in Figs. 1b and 3b. A typical PEEM image of the near-field modes excited by a 410 nm laser with circular polarization at normal incidence is shown in Fig. 1d, which indicates a dielectric vector near-field vortex mode. Additionally, two-photon photoemission is preferred for PEEM imaging at a wavelength of approximately 410 nm (Supplementary Fig. 7), that is $P_E \propto I^2$, where $P_E$ is the photoemission intensity and $I$ is the local field intensity. Compared to the SPP vortices that carry the TM mode, the low-loss dielectric near-field vortex modes are based on the fundamental TE mode, and their properties are discussed below.

The vector near-field vortex modes are excited by a circularly polarized plane beam at normal incidence (carrying spin angular moment (SAM, ±1) for left- and right-handed circular polarizations (LCP/RCP)) with an etched ring or Archimedean spiral slits as couplers (Fig. 2). Following the conservation of angular momentum and spin-orbit coupling[28,29], the topological charge of the near-field vortex is formed by the superposition of the SAM (±1) of the incident beam and

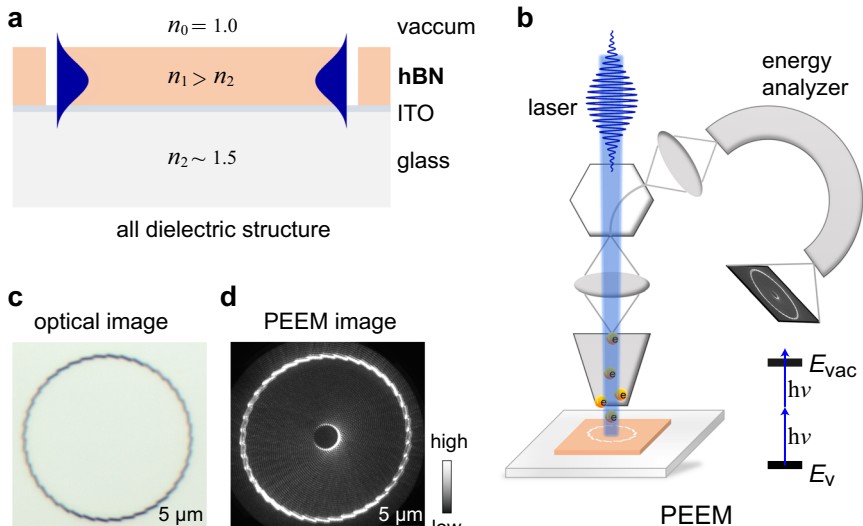

**Fig. 1 | Design of hexagonal boron nitride (hBN) nanostructures and photoemission electron microscopy (PEEM) methods. a** Schematic of a dielectric waveguide with an hBN flake on glass substrate with ~10 nm indium tin oxide (ITO) layer for avoiding surface charging in PEEM experiments. $n_0$, $n_1$ and $n_2$ denote the refractive indexes of materials. **b** Schematic of PEEM measurements with a laser

beam at normal incidence. $E_v$ and $E_{vac}$ denote the valence band and vacuum level, respectively, and h$\nu$ denote the photon energy. **c** Optical image of a typical hBN nanostructure. **d** Corresponding PEEM image excited with a 410-nm laser with right-handed circular polarization (RCP) at normal incidence.

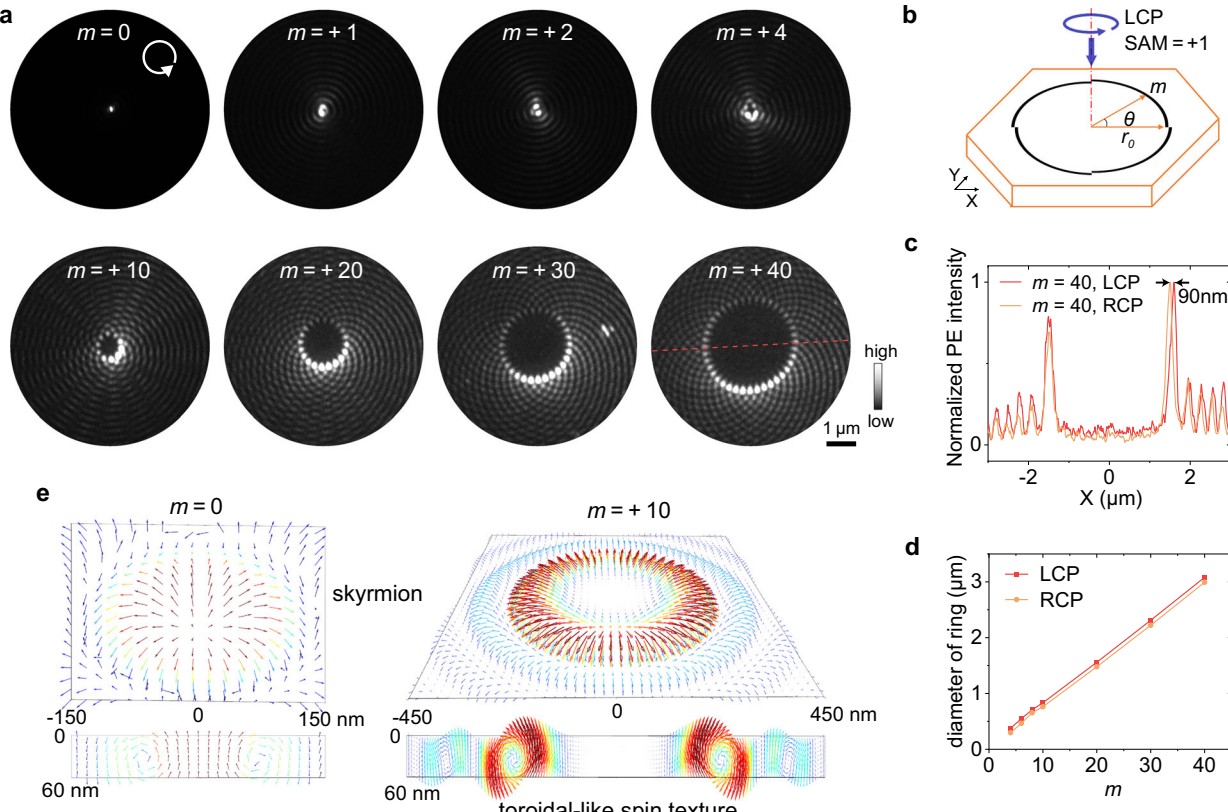

**Fig. 2 | Static dielectric near-field vortex modes measured by PEEM and inherent spin textures. a** Dielectric near-field vortex modes measured by PEEM, excited with a 410-nm plane beam with left-handed circular polarization (LCP) at normal incidence by using etched ring or Archimedean-spiral slits with different geometrical charges ($m$, $m > 0$ for left-handed rotation, $r_0 = 9.2$ μm) as the couplers. **b** Typically designed structure on hBN for the generation of near-field vortex. The $\theta$ in the image denotes the azimuthal angle. **c** Crosscut lines for the vortex ring ($m = 40$) excited with left- and right-handed circular polarizations, indicating a 90-nm difference in diameter. **d** Linear relationship between the diameter of the vortex ring and the geometrical charge ($m$). **e** Inherent optical spin textures carried by the near-field vortices, indicating a skyrmion texture at the vacuum/hBN interface with $m = 0$ and a three-dimensional toroidal-like spin texture with $m = 10$.

the geometrical charge ($\pm m$) introduced by Archimedean spiral slits[30]. To form the near-field vortices, the Archimedean spirals are designed as[10] $r = r_0 + \lambda_{eff} \cdot \mathrm{mod}(m\theta, 2\pi)/2\pi$, where $r_0$ is the initial radius, $\theta$ is the azimuthal angle, $m$ is the geometrical charge, $\lambda_{eff}$ is the effective wavelength of the TE mode defined by $\lambda_0/n_{eff}$, $\lambda_0$ is the wavelength of the incident light, and $n_{eff}$ is the effective refractive index of TE mode. For hBN thickness of ~60 nm at the excitation wavelength of 410 nm, $n_{eff}$ and $\lambda_{eff}$ are calculated as 1.78 and 230 nm, respectively, which could be well produced by PEEM experiments (Supplementary Fig. 6).

The experimental PEEM images of the near-field vortices with a series of topological charges are shown in Fig. 2a from $m = 0-40$ excited with the LCP (SAM = +1) (see also Supplementary Fig. 11 for the PEEM images excited with RCP and linear polarization). Near-field vortices were produced at slit edges and evolved into rotating vortex rings in the central region. The diameter of the vortex rings increased linearly with the topological charges, and the differences in the diameters excited with LCP/RCP were ~90 nm (Fig. 2c, d). The number of bright points surrounding the vortex ring equals the geometrical charge ($m$) because of the interference between the vortex phase singularity and the incident plane beam. The low-loss optical property of hBN plays a crucial role in the near-field intensity of the vortices, especially, a 60–80-nm nanofocusing and >$10^3$ or >$10^4$ strong photoemission enhancement could be observed with a ring slit (Supplementary Fig. 6). The <$1/4\lambda_0$ nanofocusing is attributed to the combination of the high refractive index of hBN and the two-photon process of PEEM imaging, and the size is approximately by $1/(2\sqrt{2}n_{eff})\lambda_0$.

Near-field vortices have rich inherent optical spin textures[12,31,32]. For example, the lowest-order near-field vortex has a skyrmion-like spin texture at the vacuum/hBN and hBN/glass interfaces (Fig. 2e) similar to that obtained in the SPP vortex[31,33]. Furthermore, beyond the SPP vortex, because the dielectric waveguide has a three-dimensional mode distribution, much more complex spin textures can emerge from dielectric near-field vortices. Three-dimensional toroidal-like spin textures were observed for near-field vortices with different topological charges (Fig. 2e). More details on the discussion of spin textures can be obtained in the Supplementary Note 7 (Supplementary Figs. 12–14). The structured near-field modes and vector spin textures supported by dielectric nanostructures could have potential applications in the optical control of the quantum properties of low-dimensional materials on a deep subwavelength scale with many advantages over metal structures, including low fluorescence quenching and Coulomb screening of dielectric materials, high local field intensity, and more complex structured light by dielectric modes.

## TR-PEEM measurements

Apart from the static PEEM measurements for low-loss dielectric near-field modes realized above, spatiotemporally resolved investigations could be performed with TR-PEEM through an interferometric pump-probe technique using a laser source that delivers 22 fs laser pulses with a central wavelength of 407 nm (Supplementary Fig. 2). Here, we use plane wave packet mode in dielectric waveguide to show the spatiotemporal measurements. The waveguide modes were excited from the edge coupler on hBN with laser pulses at an oblique incidence

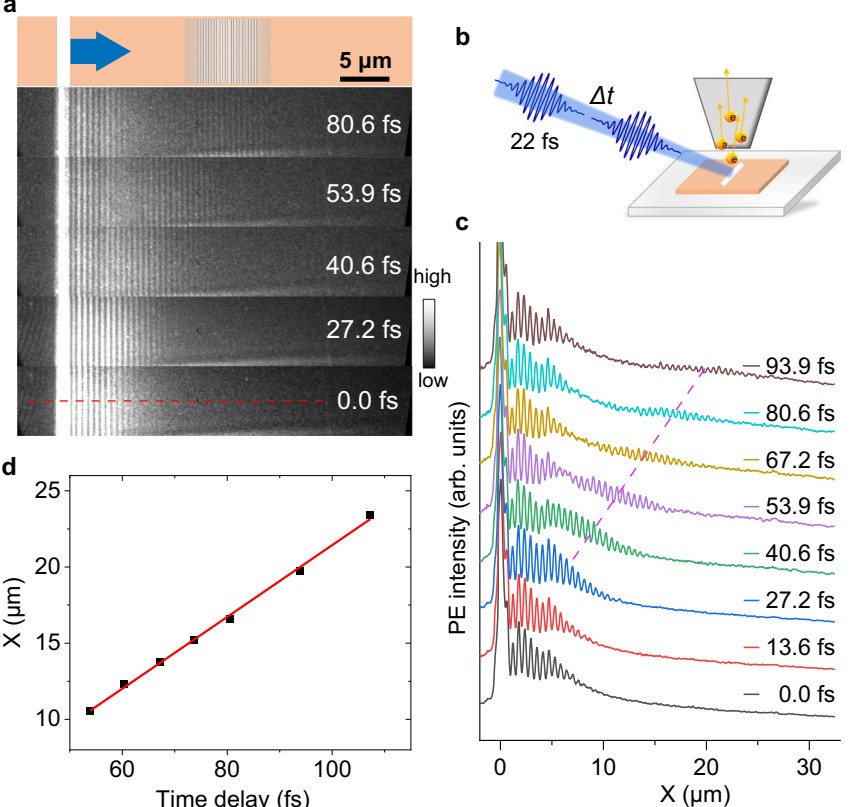

**Fig. 3 | Time-resolved PEEM (TR-PEEM) measurements of dielectric waveguide modes. a** Spatiotemporal evolution of waveguide modes measured with TR-PEEM. **b** Illustration of TR-PEEM measurement. **c** Crosscut lines from (**a**) along the red dashed line, indicating wave packet moving with time delay. The dashed pink line in (**c**) is the visual guide line. **d** Center of the observed wave packet varies with time delay, the red line is the linear fitting.

of 74° and in-plane polarization (s-polarization) (Fig. 3a, b). The PEEM images at a series of time delays between the pump and probe pulses depict the evolution and propagation of near-field modes (see also Supplementary Figs. 15 and 16). The near-field modes along the crosscut line in Fig. 3a are plotted in Fig. 3c, d for better visualization. The group velocity was extracted from the wave packets, and the group refractive index $n_g$ was calculated to be -2.2, consistent with the simulations (see Supplementary Note 9 for details). The group refractive index measured here could also be applicable to the vortex modes discussed above.

The observed wave packet is caused by the interference between waveguide mode and the second laser pulse with a time delay. Therefore, the intensity profile of laser pulse will have a significant influence on the intensity of the observed wave packet. The observed "damping" is not mainly due to the loss of hBN waveguide, but due to the small size of our laser spot. Herein, a small focusing beam spot (30 μm × 70 μm) was used in TR-PEEM measurements to increase the signals' intensity because the total laser power was small (<4 mW) in the experiments. Thus, the dark signals on the right side of the images were not due to the loss of near-field modes. By further optimizing the laser power, this technique can have promising applications in near-field investigations of complicated dielectric modes with ultrahigh spatiotemporal resolution, such as the coupling and evolution of topological edge states and cavity modes, near-field distributions and dynamics of resonant modes of metasurfaces.

### hBN photoemission source

Finally, the combination of low-loss dielectric near-field modes and the material properties of hBN can yield designs for photoelectronic devices. Herein, we propose a model for a photoemission emitter with

hBN near-field vortex modes. For the lowest-order mode, the spatial distribution of the emitter showed a nanoscale localized spot with strong intensity (Fig. 4a, b), which is the smallest lateral size (<80 nm) of flat photoemission sources, as reported previously[34,35]. It should be noted, herein, the size of the photoemission source means the working area, where the electrons are emitted, because the working area partly determines the electron coherent features. In addition, an etched ring with a large radius $r_0 = 20.7$ μm was adopted here, which indicates that the strong focusing spot in the center is an evidence for the low-loss properties of the near-field modes. The photoemission angle was evaluated by the momentum-space imaging of PEEM, indicating a sharp emission angle around the Γ point due to the atomically flat surface of hBN and the low photon energy of the excitation laser (Fig. 4c, d). The energy distribution of the emitted electrons was measured by an energy analyzer, presenting a narrow energy span of approximately 0.65 eV (Fig. 4e, f).

## Discussion

Nanoscale source size, sharp emission angle, and narrow energy spread are favorable characteristics for photoemission emitters, reflecting promising coherent features without compression[34–39]. Furthermore, the stable material properties and flat structure of hBN indicate the long lifetime of a photoemission source. Additionally, as indicated by TR-PEEM measurements for waveguide modes, hBN structures support an ultrafast temporal response with a wide spectral working range, which is appropriate for next-generation ultrafast electron sources. Additionally, the spatial mode distributions could be controlled by the polarization of laser and the topological charges of vortex modes (by changing the nanostructures or using a vortex beam as excitation light), whereas emission angle and energy distribution

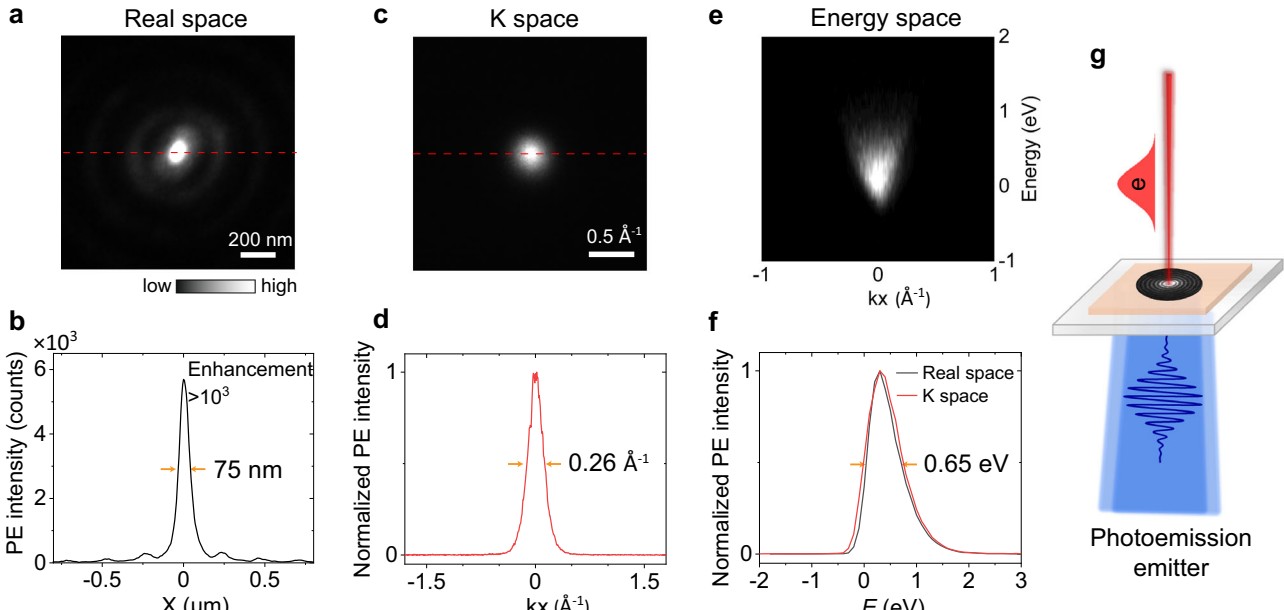

**Fig. 4 | Evaluations of photoemission properties in real, momentum, and energy spaces. a, b** Real-space PEEM image showing a nanofocusing with 75 nm size for a ring slit with $m = 0$ and large $r_0 = 20.7$ μm excited with RCP at 410 nm. **c, d** Corresponding momentum-space PEEM image without an energy filter with a size of 0.26 Å$^{-1}$. **e** Corresponding energy diagram indicating the localization of photoemitted electrons in energy and emission direction. **f** Energy distribution curves with a narrow energy spread of 0.65 eV. **g** Schematic of a design of flat photoemission emitter from back illumination with hBN near-field modes.

remained unchanged owing to the flat structure of hBN (Supplementary Fig. 18). The near-field intensity can be further improved with a circular grating coupler (Supplementary Fig. 5), leading to a nanoscale ultra-high-brightness photoemission emitter. The entire structure is composed of highly transparent materials, which facilitates the design of photoemission emitters using back-illumination with convenient light-field manipulations (Fig. 4g). The flexible control of the spatial intensity distribution of the photoemission emitter could have unique applications in 4D ultrafast electron microscopy and other ultrafast photoemission techniques[39,40].

In summary, by combining TR-PEEM with a low-dimensional wide-bandgap semiconductor material, hBN, we realized the revealing of low-loss dielectric near-field modes with high spatiotemporal resolution. Dielectric near-field vortices with topological charges up to 40 were revealed with high spatial resolution. The 90 nm difference in the diameters of the vortex rings excited by left- and right-handed circular polarizations was extracted. The <80 nm nanofocusing and strong near-field enhancement were observed with the lowest-order vortex. In addition, the spatiotemporal evolution of the waveguide modes was directly revealed using TR-PEEM, and the group velocity was obtained. Finally, by comprehensively characterizing the photoemission properties in real, momentum, and energy spaces, a photoemission emitter is proposed based on the hBN dielectric vector near-field vortex modes, and the mode distributions can be flexibly controlled by polarization and topological charges. Our work demonstrates a platform for investigating low-loss dielectric modes based on TR-PEEM, which has broad prospects in near-field investigations of dielectric topological modes and metasurfaces and in the inspection of on-chip devices.

## Methods
### Sample fabrication and characterizations
The hBN flakes were mechanically exfoliated on poly-dimethylpolysiloxane (PDMS) sheets from bulk crystals (2D semiconductors) and the thicknesses of hBN flakes were estimated by the reflected color under optical microscopy. Then, the selected hBN flakes were transferred to the soda-lime glass or silica glass substrate with ~10 nm ITO layer, by using the all-dry transfer method under

optical microscopy. Then the transferred hBN flakes were etched with a series of nanostructures by using focused ion beam (FIB) to support the desired near-field modes. All samples were annealed under an ultrahigh vacuum (approximately 10$^{-9}$ Torr) at 200 °C for 2 h before loading into the PEEM chamber. After PEEM experiments, the samples were characterized with atomic force microscopy (AFM) and scanning electron microscopy (SEM).

### PEEM measurements
The PEEM measurements were performed by using a high-resolution low-energy electron microscopy (LEEM)/PEEM system (ACSPELEEM III, Elmitec GmbH) equipped with a hemispherical electron energy analyzer. This PEEM system can support real-space, momentum-space, and energy-space measurements. The energy-resolved PEEM measurements were performed in imaging mode by inserting a 12.5-μm energy slit and sweeping the start voltage in a step of 0.1 eV to allow electrons with different energies passing through the energy slit. A commercial Ti:sapphire femtosecond laser (Mai Tai HP, Spectra-Physics, pulse duration: 80–100 fs) operated at 690–1040 nm (2.9 W @ 820 nm) with a repetition rate of 80 MHz was used as the laser source to pump an optical parametric oscillator (OPO) (Inspire Auto 100, SpectraPhysics). The second harmonic generation (SHG) port (390–440 nm, 1.0 W @ 410 nm) was used to provide excitation laser for the static PEEM measurements. The pulses were focused on the sample surface at normal incidence with a spot diameter of approximately 150 μm. The polarization of laser is adjusted through a 1/4 waveplate or 1/2 waveplate. For time-resolved PEEM (TR-PEEM) measurements, a laser source (Venteon CEP5, Laser Quantum) with a pulse duration of approximately 6.5 fs, a spectrum covering 620–1050 nm and a repetition rate of 80 MHz, was used to produce SHG pulses through a thin BBO crystal. The SHG pulses were used for TR-PEEM measurements at oblique incidence. The pulse duration was evaluated to be about 22 fs and the spectrum was measured with a central wavelength of 407 nm and a full width at half maximum (FWHM) of 15 nm. The power of the SHG pulses was measured to be only 4 mW, as a result of the thin thickness of BBO crystal and the relatively low output power of the laser source (<280 mW).

## Data availability

The data supporting the findings are displayed in the main text and the Supplementary Information. All raw data are available from the corresponding authors upon request.

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

## Acknowledgements

This work was supported by the following grants: National Key Research and Development Program of China under Grant Nos. 2018YFB2200403 (X.H.), 2018YFA0704404 (X.H.), 2018YFA0306302 (Y.G.). Guangdong Major Project of Basic and Applied Basic Research under Grant No. 2020B0301030009 (Q.G.). National Natural Science Foundation of China under Grant Nos. 91950204 (X.H.), 92150302 (X.H.), 92250305 (G.L.). National Postdoctoral Program for Innovative Talents under Grant No. 8206200074 (Y.L.). The Fundamental Research Funds for the Central Universities (Y.G.).

## Author contributions

Y.L., Y.G., and X.H. conceived the idea. Y.L., P.J., and H.Q. conducted simulations and data analyses. Y.L., P.J., X. Lyu, X. Li, J.T., and Z.X. conducted the experiments. Q.S., G.L., H.Y., and Q.G. contributed to writing the original draft, reviewing, and editing.

## Competing interests

The authors declare no competing interests.
