## [Peer Review File · Nature Communications]

Revealing low-loss dielectric near-field modes of hexagonal boron nitride by photoemission electron microscopyREVIEWER COMMENTS

Reviewer #1 (Remarks to the Author):

The authors report on time-integrated and time-resolved PEEM measurements of different Archimedean spiral structures with varying geometrical phase in h-BN films. They observe nanofocusing and study the dependence on the total orbital angular momentum, composed of the handedness of the laser light and the geometrical phase OAM. The results are pretty much similar to the ones reported in their references 10-12, apart from the fact that they use h-BN instead of gold.

So does this warrant a publication in Nat. Comm.? I believe not, as the key physics is contained in the topological aspects of the work, which have well been laid out in refs 10-12 of the authors.

It is a material aspect that is in my humble opinion the only difference to the previous work.

Now does this material aspect make a difference that would warrant publication in Nat. Comm.

and makes it so novel, exciting, different that it would fulfill the criteria of this journal? Again, the answer is no.

Reviewer #2 (Remarks to the Author):

The authors demonstrated the photoemission electron microscope studies of insulating hBN flakes. Various experiments were conducted, including transverse electric wave guide vortex mode along with the associated spin textures, propagating modes with their spatiotemporal dynamics, and the photoemission properties in the k-space near the gamma point. The idea of studying hBN is strongly needed in photoemission communities, particularly on the nanoscale because these thin flakes tend to be on the micron size. While the experimental data were nicely presented, I think the discussion of the underlying physics of the reported observations should be improved. In addition, the connection among vortex mode, plane wave mode, and the k-space photoemission should be strengthened. Currently, they seem to be three separated topics without in-depth investigation. Only before these parts are improved can I recommend it for publication. Major points are listed below:

1. Sample charging was mentioned as the most challenging part in doing photoemission measurements on non-conductive samples, and the author used 10nm ITO to nicely reduce the charge and claimed this as a major breakthrough. The mechanism for such improvement, however, was not discussed at all; the answer to such question is essential to future photoemission studies. For example, can the authors discuss what makes ITO special in reducing charging, why not other thin metallic films? What is the electron mean free path in hBN, is it long enough (>60-80 nm) to reach to the top surface for photoemission? What is the charging effect as a function of the hBN thickness, excitation density etc.
2. In terms of the spin textures, the skyrmion-like plasmonic spin textures are due to the TM nature of plasmon waves. In the reported vortex mode, It is not clear how the TE mode would affect the textures at the center compared to TM mode. A detailed discussion of how the vectorial electromagnetic fields form in dielectric materials, their phase distributions, in addition to the pointing vector shown in the SI, will be appreciated.
3. As to the spatiotemporal dynamics of the TE waveguide mode, it appears to have a strong damping as it propagates, which is unlike a low-loss photonic mode as the author introduced. What is the mechanism for such damping? How does the presence of ITO affect the guided mode? Does the propagation velocity match the expected photonic mode?
4. Lastly, the authors claimed near-vertical photoemission, but never discuss the physics of such term. The only thing shown is the photoemission near the gamma point, which is a limit due to the insufficient excitation energy. Can the authors clarify where do these photoelectrons come from based on the band structure of hBN? Are these purely secondary electrons? What momentum distribution does one expect for such excitation (not limited to the PE process).

Reviewer #3 (Remarks to the Author):

In this manuscript, the authors have successfully imaged the propagation of electromagnetic surface waves in low-loss dielectric materials, specifically hBN 2D semiconductor, utilizing photoemission electron microscopy technique with high spatial and temporal resolution. By employing an Archimedean spiral slit, the authors visualized orbital angular momentum states with topological charges up to $m = 40$. Furthermore, using a ring slit, they experimentally demonstrated the ability to create localized and strong photoemission effects at the vortices. While similar works have been previously demonstrated on surface plasmon measurements on metal surfaces, the novelty of this study lies in its adaptation of the technique to dielectric materials, which holds significant importance for optical device applications. The experimental results presented in this paper are sound. However, it is worth noting that certain sections of the manuscript could benefit from further proofreading and polishing of the written English. If the authors can thoroughly proofread the paper and address the following questions/comments, I would recommend publication in Nature Communications.

1. In the opening sentence of the abstract, the author said "Low-loss dielectric modes are fundamental components...", this is confusing, as the low-loss modes are not an actual physical component, they are features or phenomenon that is important for any optical components in an optical devices. The author should clarify that.
2. In the abstract, it is not clear to me how low-loss and atomically flat hBN is related to "strong nanofocusing in real space, near-vertical photoemission in momentum space, and narrow spread in energy space" (in Line 24 and 25). After reading through the whole paper, I understand that this is the related to the waveguide structure. Please distinguish clearly what are the causes and what are the effects.
3. In figure 1, if the sample is in an electron microscope, I assume instead of air, the top interface should be in vacuum? Or the authors are indeed studying devices with top air interface?
4. In line 79, the authors mention that low optical absorption is maintained in the visible to uv range, can the author clarify what is the meaning of "low"?
5. One of the key point in this paper is about imaging the electromagnetic waves in hBN. The authors pointed out correctly that it is challenging to image hBN which is an insulator with a large bandgap in the PEEM due to issue with surface charging, can the authors explain in more details how this problem is avoided? How careful selection of materials to balance the bandgap and surface charging work? (Line 53)
6. Can the authors explain how low-loss dielectric is related to strong photoemission enhancement? How much more enhancement do we gain when we compare a hBN waveguide to a normal metal waveguide of the same design? Can the authors use PEEM to measure the loss is happening in the waveguide, for example from the decrease of photoemission intensity during propagation?
7. What are the images in S5a and b. Are the left and right images identical with the right one intensity enhanced?
8. In line 189, the author claims to have created the smallest photoemission source, what is its size? In line 190, the authors mentioned that a radius of $r_0=20.7\mu\text{m}$ is adopted here. It is very confusing. The author should clarify that this radius is the size of the waveguide.
9. In the SI, line 98. Can the authors explain the choice of slit width, are all the simulation done with the slit width of 180nm and how does it affect the simulation?

We thank the reviewers for their careful reviewing of our manuscript and for their detailed and constructive reports. During the revision, we have taken all the comments from the reviewers into consideration. Our responses to each of the reviewers' comments are as follows and highlighted in blue.

Report of Reviewer #1

The authors report on time-integrated and time-resolved PEEM measurements of different Archimedean spiral structures with varying geometrical phase in h-BN films. They observe nanofocusing and study the dependence on the total orbital angular momentum, composed of the handedness of the laser light and the geometrical phase OAM. The results are pretty much similar to the ones reported in their references 10-12, apart from the fact that they use h-BN instead of gold. So does this warrant a publication in Nat. Comm.? I believe not, as the key physics is contained in the topological aspects of the work, which have well been laid out in refs 10-12 of the authors. It is a material aspect that is in my humble opinion the only difference to the previous work. Now does this material aspect make a difference that would warrant publication in Nat. Comm. and makes it so novel, exciting, different that it would fulfill the criteria of this journal? Again, the answer is no.

Reply:

We thank the referee for the comments. However, we disagree with the referee's opinion.

Firstly, we make a breakthrough in the application of TR-PEEM technique for the investigation of dielectric near-field modes. As we demonstrate in the manuscript, although TR-PEEM has been successfully applied to investigate near-field modes supported by metal for many years, the extension to dielectric modes are still not very successful and we find a solution to this problem by using hBN. The static vortex modes and the dynamic moving of waveguide packets are convenient examples to show the successful observation of dielectric near-field modes. Moreover, the vortex modes supported by hBN waveguide have many characteristics different from SPP. Apart from

TE vs TM mode, and low loss vs high loss, the major difference in topology is caused by the dimension. The dielectric waveguide modes have a three-dimensional distribution while the SPP has a two-dimensional distribution, making the dielectric waveguides have more complex spin textures and could be explored further in the future. In addition, this technique has broad prospects in the investigations of dielectric topological edge states, near-field modes of metasurfaces, and in the inspection of on-chip devices.

Secondly, the dielectric near-field modes have much wider applications than SPP due to their much lower loss and can realize much stronger near-field enhancement, as already demonstrated in the manuscript, including optical control of the quantum properties of low-dimensional materials on a deep subwavelength scale, enhancement of high harmonic generation, etc.

Finally, hBN itself, as a novel van der Waals material, is worth investigations in the optical and electronic properties and could bring surprises in interdisciplinary applications. The hBN photoemission source proposed in the manuscript is a good example by combing the advantages of nanophotonics and new materials. In summary, we believe this work will evoke broad interest in many fields, including TR-PEEM technique, dielectric nanophotonics, novel materials and devices.

Report of Reviewer #2

The authors demonstrated the photoemission electron microscope studies of insulating hBN flakes. Various experiments were conducted, including transverse electric waveguide vortex mode along with the associated spin textures, propagating modes with their spatiotemporal dynamics, and the photoemission properties in the k-space near the gamma point. The idea of studying hBN is strongly needed in photoemission communities, particularly on the nanoscale because these thin flakes tend to be on the micron size. While the experimental data were nicely presented, I think the discussion of the underlying physics of the reported observations should be improved. In addition, the connection among vortex mode, plane wave mode, and the k-space photoemission should be strengthened. Currently, they seem to be three separated topics without in-

depth investigation. Only before these parts are improved can I recommend it for publication. Major points are listed below:

Reply:

We thank the referee for the helpful comments. We agree with the referee that “The idea of studying hBN is strongly needed in photoemission communities, particularly on the nanoscale because these thin flakes tend to be on the micron size.” Inspired from this comment, we think that the hBN nanoscale modes could also have potential application in TR-ARPES to produce nanoscale pump laser excitation. We also thank the referee for the suggestions on discussions of the underlying physics and the arrangement of paragraphs, and we make the corresponding modifications.

Comment 1. Sample charging was mentioned as the most challenging part in doing photoemission measurements on non-conductive samples, and the author used 10nm ITO to nicely reduce the charge and claimed this as a major breakthrough. The mechanism for such improvement, however, was not discussed at all; the answer to such question is essential to future photoemission studies. For example, can the authors discuss what makes ITO special in reducing charging, why not other thin metallic films? What is the electron mean free path in hBN, is it long enough (>60-80 nm) to reach to the top surface for photoemission? What is the charging effect as a function of the hBN thickness, excitation density etc.

Reply:

We thank the referee for the helpful comments. As the referee mentioned, in this work, we use 10 nm ITO as the conductive layer rather than other metallic film. The main consideration is about the loss caused by optical absorption. ITO is transparent in visible range, which means it has much lower absorption than metallic film, such as gold and monolayer graphene. The 10 nm ITO has little absorption and therefore has very weak influence to the waveguide modes in hBN. It should be noted, the thickness of ITO is chosen to be very thin rather than commonly used 100 nm or 150 nm, because ITO itself still has some absorption although small. Therefore, the thickness down to

10 nm is preferred to maintain the low-loss hBN waveguide modes. The simulations for the influence of ITO thickness on the intensity of hBN waveguide mode are shown in Fig. R1.

Fig. R1. Simulated decay of waveguide mode in hBN along propagation direction with different ITO layer thicknesses: 0 nm, 10 nm and 50 nm. The hBN waveguide with 2 nm Au as conductive layer is also plotted for comparison. The waveguide mode is excited from a line slit on 60 nm hBN with 410 nm laser at normal incidence.

To clarify the choice of ITO, we add a discussion in the supplementary material: “It should be noted, a very thin ITO down to 10 nm is adopted in this work as conductive layer in order to avoid introducing absorption loss. ITO has much lower absorption loss than thin metallic film, such as gold and monolayer graphene. However, ITO itself still has a small absorption, therefore, the thickness down to 10 nm is preferred to maintain the low-loss hBN waveguide modes. The simulations for the influence of ITO thickness on the intensity of hBN waveguide mode are shown in Fig. S8.”

In addition, herein, to directly visualize the advantage of using thin ITO, we give the PEEM image of hBN ring covered with monolayer MoSe₂, as shown in Fig. R2. As we can see, the absorption of monolayer MoSe₂ could bring a catastrophic influence on the nanofocusing intensity of hBN ring.

Fig. R2. PEEM image for hBN waveguide covered with monolayer MoSe₂ excited with 410 nm laser at normal incidence with vertical polarization. A severe intensity decay can be observed due to the absorption of monolayer MoSe₂, which indicates that monolayer MoSe₂ even has much larger absorption than 10 nm ITO.

For the electron mean free path in hBN, as we know, the electron mean free path is on the order of 1~2 nm or less (P. Sutter and E. Sutter, *APL Mater.*, 2, 092502 (2014); Pablo de Vera and Rafael Garcia-Molina, *J. Phys. Chem. C*, 123,2075-2083 (2019)). There could be some misunderstanding in asking the question “is it long enough (>60-80 nm) to reach to the top surface for photoemission?” Because the photoemission on hBN is directly from the surface layer (within 2 nm) of hBN, not from the ITO under the hBN. That is to say, the electron mean free path has no direct connection with the thickness of hBN. For the charging effect, the hBN thickness up to 80 nm has no obvious charging phenomenon in our study, because charges can be conducted through the underling ITO layer. In contrast, if the hBN and underling ITO are etched with a deep ring slit to be isolated from the surrounding ITO substrate, charging effect will appear easily, which means the ITO layer is very important in avoiding charging. In addition, as already shown in the supplementary material Fig. S7b, the excitation density could have some influence on charging effect, because we observed that when the laser power was above 300 mW, the photoemission intensity became saturated. This saturation effect could be attributed to the limited charge transfer efficiency from hBN to ITO.

To clarify the charging effect, we add some discussions in the supplementary

material: “The electron mean free path of hBN is on the order of 1~2 nm or less. The photoemission on hBN is directly from the surface layer (within 2 nm) of hBN, not from the ITO under the hBN, which means the electron mean free path has no direct connection with the thickness of hBN. For the charging effect, the hBN thickness up to 80 nm has no obvious charging phenomenon in our study, because charges can be conducted through the underling ITO layer. In contrast, if the hBN and underling ITO are etched with a deep ring slit to be isolated from the surrounding ITO substrate, charging effect will appear easily, which means the ITO layer is very important in avoiding charging. In addition, as shown in Fig. S7b, the excitation density could have some influence on charging effect, because we observed that when the laser power was above 300 mW, the photoemission intensity became saturated. This saturation effect could be attributed to the limited charge transfer efficiency from hBN to ITO.”

Comment 2. In terms of the spin textures, the skyrmion-like plasmonic spin textures are due to the TM nature of plasmon waves. In the reported vortex mode, It is not clear how the TE mode would affect the textures at the center compared to TM mode. A detailed discussion of how the vectorial electromagnetic fields form in dielectric materials, their phase distributions, in addition to the pointing vector shown in the SI, will be appreciated.

Reply:

We thank the referee for the helpful suggestions. We agree that the difference of TE mode supported by hBN waveguide and TM mode supported by SPP should be discussed more detailly.

Firstly, as we all know, the electric and magnetic field components are correlated with each other in an electromagnetic field. The magnetic field component in the fundamental TE mode is similar or corresponding to the electric field component in the TM mode. That is to say, the magnetic field component in TE mode of hBN waveguide will have similar distribution with the electric field component of SPP. Considering this correlation, the cycle-averaged Poynting vector $\mathbf{P} = \text{Re}(\mathbf{E}^* \times \mathbf{H})/2$ of TE mode of hBN will be similar to that of TM mode of SPP, and the SAM has an inherent correlation with Poynting vector. That is why TE mode supported by hBN waveguide

at an interface has the similar SAM texture as TM mode supported by SPP.

Secondly, from the point of vector electromagnetic fields, taking the lowest vortex as an example, the TE mode is selectively excited from the ring slit by circularly polarized light, resulting in a phase retardation, similarly as that in TM mode of SPP. The phase retardation causes the formation of near-field vortex, and the SAM texture is in fact the inherent characteristic of the vortex. The vortex is a vector near-field vortex and can be decomposed into the scalar vortex with LCP/RCP as the bases. As shown in Fig. S12, the total vector electric fields are decomposed to LCP/RCP components. The LCP component carries no vortex topological charge and forms a focusing spot in the center, while the RCP component carries a vortex topological charge of 2 and therefore forms a doughnut field distribution with a rotating phase. It should be noted, the LCP and RCP components have opposite spin orientation, therefore, the nesting of LCP/RCP components results in the spin-flipping structure, that is SAM texture.

To emphasize the discussion, we add a paragraph in the supplementary material: “The difference of TE mode supported by hBN waveguide and TM mode supported by SPP should be discussed more detailly. Firstly, as we all know, the electric and magnetic field components are correlated with each other in an electromagnetic field. The magnetic field component in the fundamental TE mode is similar or corresponding to the electric field component in the TM mode. That is to say, the magnetic field component in TE mode of hBN waveguide will have similar distribution with the electric field component of SPP. Considering this correlation, the cycle-averaged Poynting vector $\mathbf{P} = \text{Re}(\mathbf{E}^* \times \mathbf{H})/2$ of TE mode of hBN will be similar to that of TM mode of SPP, and the SAM has an inherent correlation with Poynting vector. That is why TE mode supported by hBN waveguide at an interface has the similar SAM texture as TM mode supported by SPP.

Secondly, from the point of vector electromagnetic fields, taking the lowest vortex as an example, the TE mode is selectively excited from the ring slit by circularly polarized light, resulting in a phase retardation, similarly as that in TM mode of SPP. The phase retardation causes the formation of near-field vortex, and the SAM texture is in fact the inherent characteristic of the vortex. The vortex is a vector near-field vortex

and can be decomposed into the scalar vortex with LCP/RCP as the bases. As shown in Fig. S12, the total vector electric fields are decomposed to LCP/RCP components, the LCP component carries no vortex topological charge and forms a focusing spot in the center, while the RCP component carries a vortex topological charge of 2 and therefore forms a doughnut field distribution with a rotating phase. It should be noted, the LCP and RCP components have opposite spin orientation, therefore, the nesting of LCP/RCP components results in the spin-flipping structure, that is SAM texture.”

Comment 3. As to the spatiotemporal dynamics of the TE waveguide mode, it appears to have a strong damping as it propagates, which is unlike a low-loss photonic mode as the author introduced. What is the mechanism for such damping? How does the presence of ITO affect the guided mode? Does the propagation velocity match the expected photonic mode?

Reply:

We thank the referee for the helpful comments. As we mentioned in the main manuscript, the observed “damping” is not due to the loss of hBN waveguide, but due to the limited size of our laser spot. The laser profile has a gradually weak and weak intensity distribution from the center to side, especially on the right side of the PEEM image in Fig. 3. The observed wave packet is caused by the interference between waveguide mode and the second laser pulse with a time delay. Therefore, the intensity profile of laser pulse will have a significant influence on the intensity of the observed wave packet. On the contrary, the 10 nm ITO has a weak influence on the damping, as already discussed in Comment 1.

For the propagation velocity, as already shown in the manuscript, the phase velocity extracted from PEEM measurements is consistent with the FDTD simulations. Here, we give the simulated group velocity, as shown in Fig. R3. The wave packet moves from position at 3 μm to 8 μm with 36.7 fs. Then, the group velocity is calculated to be 0.137 $\mu\text{m}/\text{fs}$, and the corresponding group refractive index $n_g = 2.2$, well consistent with the experimental result.

Fig. R3. Simulated wave packet moving from position at 3 μm to 8 μm with 36.7 fs. The group velocity is calculated to be 0.137 $\mu\text{m}/\text{fs}$, and the corresponding group refractive index $n_g = 2.2$, well consistent with the experimental result.

To emphasize the referee’s comments, we add a discussion in the main manuscript: “The observed wave packet is the interference between waveguide mode and the second laser pulse with a time delay. Therefore, the intensity profile of laser pulse will have a significant influence on the intensity of the observed wave packet. The observed “damping” is not mainly due to the loss of hBN waveguide, but due to the small size of our laser spot”. We also add a sentence “consistent with the simulations” after the experimental group velocity mentioned in main manuscript and add the simulation results in the supplementary material.

Comment 4. Lastly, the authors claimed near-vertical photoemission, but never discuss the physics of such term. The only thing shown is the photoemission near the gamma point, which is a limit due to the insufficient excitation energy. Can the authors clarify where do these photoelectrons come from based on the band structure of hBN? Are these purely secondary electrons? What momentum distribution does one expect for such excitation (not limited to the PE process).

Reply:

We thank the referee for the helpful suggestions. It’s true that we must consider the band structure of hBN to understand the photoemission physics with 410 nm laser.

Firstly, the photoemission of hBN with 410 nm laser is a two-photon process, as shown in the power-dependent measurements given in the supplementary material. That is to say, two photons (2×3.02 eV) are absorbed to overcome the work function of hBN. Considering the large bandgap (~ 6 eV) of hBN, the interband excitation from

valence band to conduction band and the following valley scattering are not expected. The photoemission is expected to be directly excited from the valence band around Γ point of hBN band structure to vacuum, overcoming the work function (The band structure is not shown here, because it can be easily found from literatures). Considering the two photons' energy ($2 \times 3.02 = 6.04$ eV), to fill the energy conservation and in-plane momentum conservation during photoemission, the momentum space of photoemission is very small and localized around Γ point, that is why the photoemission is near vertical.

In addition, considering above discussion and power-dependent measurements, the photoemitted electrons are not from secondary electrons, but attributed to direct two-photon photoemission.

Finally, we want to provide an extra discussion on the photoemission process that not pointed out by the referee. The strong nanofocusing on hBN causes nanoscale strong light field, which could induce defects at the focusing region on the surface of hBN. The defects could introduce defect states and provide electron density of states in the bandgap, which could work as intermediate energy levels for two-photon photoemission. With the assistance of the intermediate energy levels, the two-photon photoemission could be considerably enhanced, which was also observed in our PEEM experiments. We found that the photoemission could be enhanced under laser illumination in several hours, which could be attributed to laser induced interband defect states.

To clarify the argument, we add a discussion in the supplementary material: "The photoemission of hBN with 410 nm laser is a two-photon process, as shown in the power-dependent measurements. Two photons (2×3.02 eV) are absorbed to overcome the work function of hBN. Considering the large bandgap (~ 6 eV) of hBN, the interband excitation from valence band to conduction band and the following valley scattering are not expected. The photoemission is expected to be directly excited from the valence band around Γ point of hBN band structure to vacuum, overcoming the work function. Considering the two photons' energy ($2 \times 3.02 = 6.04$ eV), to fill the energy conservation and in-plane momentum conservation during photoemission, the momentum space of

photoemission is very small and localized around Γ point, that is why the photoemission is near vertical. In addition, considering above discussion and power-dependent measurements, the photoemitted electrons are not from secondary electrons, but attributed to direct two-photon photoemission”.

Report of Reviewer #3

In this manuscript, the authors have successfully imaged the propagation of electromagnetic surface waves in low-loss dielectric materials, specifically hBN 2D semiconductor, utilizing photoemission electron microscopy technique with high spatial and temporal resolution. By employing an Archimedean spiral slit, the authors visualized orbital angular momentum states with topological charges up to $m = 40$. Furthermore, using a ring slit, they experimentally demonstrated the ability to create localized and strong photoemission effects at the vortices. While similar works have been previously demonstrated on surface plasmon measurements on metal surfaces, the novelty of this study lies in its adaptation of the technique to dielectric materials, which holds significant importance for optical device applications. The experimental results presented in this paper are sound. However, it is worth noting that certain sections of the manuscript could benefit from further proofreading and polishing of the written English. If the authors can thoroughly proofread the paper and address the following questions/comments, I would recommend publication in Nature Communications.

Reply:

We thank the referee for the helpful comments. We agree with the referee that “The novelty of this study lies in its adaptation of the technique to dielectric materials, which holds significant importance for optical device applications.” We further proofread and polish the manuscript following the referee’s suggestions.

Comment 1. In the opening sentence of the abstract, the author said “Low-loss dielectric modes are fundamental components....”, this is confusing, as the low-loss modes are not an actual physical component, they are features or phenomenon that is important for any optical components in an optical devices. The author should clarify that.

Reply:

We thank the referee for the helpful suggestions. We accept the referee's suggestions and revised the corresponding sentence as "Low-loss dielectric modes are important features and functional bases of fundamental optical components in on-chip devices". In addition, following this suggestion, we revised the first sentence of the main text as "Low-loss dielectric modes such as dielectric waveguide modes, topological edge states, metasurfaces and whispering-gallery modes are important features and functional bases of fundamental optical components for constructing on-chip optical devices".

Comment 2. In the abstract, it is not clear to me how low-loss and atomically flat hBN is related to "strong nanofocusing in real space, near-vertical photoemission in momentum space, and narrow spread in energy space" (in Line 24 and 25). After reading through the whole paper, I understand that this is related to the waveguide structure. Please distinguish clearly what are the causes and what are the effects.

Reply:

We thank the referee for the helpful suggestions. We accept the referee's suggestions and revised the corresponding sentence as "With the lowest-order vortex structure, strong nanofocusing in real space is realized, while near-vertical photoemission in momentum space, and narrow spread in energy space are simultaneously achieved due to the atomically flat surface of hBN and the two-photon photoemission process, providing a novel scheme for flat photoemission emitters".

Comment 3. In figure 1, if the sample is in an electron microscope, I assume instead of air, the top interface should be in vacuum? Or the authors are indeed studying devices with top air interface?

Reply:

We thank the referee for the helpful suggestions. We accept the referee's suggestions and revised "air" to "vacuum" in the whole manuscript.

Comment 4. In line 79, the authors mention that low optical absorption is maintained

in the visible to uv range, can the author clarify what is the meaning of “low”?

Reply:

We thank the referee for the helpful suggestions. We accept the referee’s suggestions and revised the corresponding sentence to “Because hBN is a wide-bandgap semiconductor, theoretically, it has no optical absorption in the visible-to-ultraviolet range and is suitable for constructing low-loss nanophotonic structures”.

Comment 5. One of the key point in this paper is about imaging the electromagnetic waves in hBN. The authors pointed out correctly that it is challenging to image hBN which is an insulator with a large bandgap in the PEEM due to issue with surface charging, can the authors explain in more details how this problem is avoided? How careful selection of materials to balance the bandgap and surface charging work? (Line 53)

Reply:

We thank the referee for the helpful suggestions. In this study, the surface charging problem is avoided by using a thin ITO layer, which conducts charges away from the hBN flakes. In our opinion, the hBN used in this study is relatively small flakes (in-plane size <300 μm), rather than film that covering the whole surface, which could also be an advantage for transferring charges from hBN to underlying ITO. In addition, the hBN flake can be clearly imaged by PEEM without any blur. For example, the hBN flakes can also be clearly imaged by PEEM with Si substrate.

The careful selection of materials to balance the bandgap and surface charging is needed because bandgap and conductivity is contradictory, large bandgap generally means low conductivity. To construct optical devices in visible range, large bandgap is preferred. However, common optical materials such as Si_3N_4 , LiNbO_3 are not conductive, thus cannot be measured in PEEM. The materials with smaller bandgap such as Si, GaAs have sufficient conductivity, but they are not suitable for visible range due to large optical absorption. Therefore, the selection of materials is an important task. It could be a good idea to seek among novel materials. The van der Waals materials

could be potential choices as they have special layered structure. And through the PEEM experiments, we found hBN is compatible with PEEM. It should be noted, it's hard to predict if the materials are suitable for PEEM measurements, the experimental attempts should be required.

To clarify this problem, we add the discussion above in the supplementary material: "To note, the hBN used in this study is relatively small flakes (in-plane size $<300\ \mu\text{m}$), rather than film that covering the whole surface, which could also be an advantage for transferring charges from hBN to underlying ITO. In addition, the hBN flake can be clearly imaged by PEEM without any blur. For example, the hBN flakes can also be clearly imaged by PEEM with Si substrate.

The careful selection of materials to balance the bandgap and surface charging is needed because bandgap and conductivity is contradictory to some extent, large bandgap generally means low conductivity. To construct optical devices in visible range, large bandgap is preferred. However, common optical materials such as Si_3N_4 , LiNbO_3 are not conductive, thus cannot be measured in PEEM. The materials with smaller bandgap such as Si, GaAs have sufficient conductivity, but they are not suitable for visible range due to large optical absorption. Therefore, the selection of materials is an important task. It could be a good idea to seek among novel materials. The van der Waals materials could be potential choices as they have special layered structure. And through the PEEM experiments, we found hBN is compatible with PEEM. It should be noted, it's hard to predict if the materials are suitable for PEEM measurements, the experimental attempts should be required."

Comment 6. Can the authors explain how low-loss dielectric is related to strong photoemission enhancement? How much more enhancement do we gain when we compare a hBN waveguide to a normal metal waveguide of the same design? Can the authors use PEEM to measure the loss is happening in the waveguide, for example from the decrease of photoemission intensity during propagation?

Reply:

We thank the referee for the helpful comments. As demonstrated in the manuscript,

the low loss of dielectric material is a prerequisite for strong photoemission enhancement. The enhancement is achieved by focusing the waveguide mode excited from the ring slit edge. That's to say, the waveguide mode is coupled into the hBN slab from the slit edge and propagates toward the center to form a strong nanofocusing. Therefore, the loss during propagation will have a vital influence on the final focusing intensity. From the simulations, without considering loss, the focusing intensity increases with the ring diameter, because more light is collected by the slit edge with a larger circumference. And the focusing intensity can be further largely enhanced by using circular grating coupler. As for SPP supported by metal, large loss is expected, resulting in weak enhancement. And it's expected that the focusing intensity cannot be efficiently enhanced by increasing the diameter due to the propagation loss of SPP. As shown in Fig. R4, by using the same design, the intensity of focusing spot with hBN waveguide is much larger than that with gold film. The intensity $|\mathbf{E}|^2$ at vacuum/hBN is ~ 12 times larger than that at vacuum/Au. In addition, the maximum intensity of hBN waveguide is not at the surface, but inside hBN, as shown in Fig. R4a, which is ~ 23 times larger than that at vacuum/Au. It should be noted, for SPP, the ring slit ($m=0$) excited with LCP does not form a focusing point in the center, but a doughnut pattern. To form a focusing point, spiral slit ($m=-1$) should be adopted and we use the focusing point to compare the intensity.

Fig. R4. Comparison of the nanofocusing intensities with hBN waveguide and SPP. (a) Intensity $|\mathbf{E}|^2$ for hBN waveguide excited with 410 nm LCP laser on a ring slit (radius $7.36 \mu\text{m}$ ($32\lambda_{\text{eff}}$), slit width 180 nm, hBN thickness 60 nm). **(b)** Intensity $|\mathbf{E}|^2$ for Au SPP excited with

800 nm LCP laser on a spiral slit ($m=-1$) (initial radius $7.02 \mu\text{m}$ ($9\lambda_{\text{eff}}$), slit width 200 nm, Au thickness 60 nm). (c) Intensity $|E|^2$ for Au SPP excited with 800 nm LCP laser on a ring slit (radius $7.02 \mu\text{m}$ ($9\lambda_{\text{eff}}$), slit width 200 nm, Au thickness 60 nm).

In addition, we can try to measure the loss from the decrease of photoemission intensity during propagation. As shown in Fig. R5a,b, we measured the photoemission intensity along the propagation direction excited from a slit with 410 nm laser at normal incidence. The decay of photoemission intensity along the propagation direction can be observed. Considering the two-photon photoemission process, $P_E \propto I^2 \propto |E|^4$, the electric field $|E|$ along the propagation direction is extracted in Fig. R5c. From $X = 2 \mu\text{m}$ to $X = 7 \mu\text{m}$, the amplitude of $|E|$ (between the red dashed lines) decreases from 0.97 to 0.77, which means the amplitude of $|E|$ has a 20% decrease in $5 \mu\text{m}$. In contrast, in simulations, the amplitude of $|E|$ has a 7.5% decrease in $5 \mu\text{m}$. The larger loss measured in experiments could be due to fabrication quality and the inhomogeneous laser spot. Because the relatively small laser spot ($\sim 150 \mu\text{m}$) was used in normal incidence, it's hard to evaluate the loss accurately, and the measured value should be larger than actual loss.

Fig. R5. (a) PEEM image for line slit excited with TE polarization with 410 nm laser at normal incidence. (b) crosscut line from (a). (c) Extracted amplitude of $|E|$ following $P_E \propto I^2 \propto |E|^4$, the amplitude of $|E|$ has a 20% decrease in $5 \mu\text{m}$.

To clarify this issue, we add some discussions in the supplementary material:

“The low loss of dielectric material is a prerequisite for strong photoemission enhancement. The enhancement is achieved by focusing the waveguide mode excited from the ring slit edge. That’s to say, the waveguide mode is coupled into the hBN slab from the slit edge and propagates toward the center to form a strong nanofocusing.

Therefore, the loss during propagation will have a vital influence on the final focusing intensity. From the simulations, without considering loss, the focusing intensity increases with the ring diameter, because more light is collected by the slit edge with a larger circumference. And the focusing intensity can be further largely enhanced by using circular grating coupler. As for SPP supported by metal, large loss is expected, resulting in weak enhancement. And it's expected that the focusing intensity cannot be efficiently enhanced by increasing the diameter due to the propagation loss of SPP. As shown in Fig. S9, by using the same design, the intensity of focusing spot with hBN waveguide is much larger than that with gold film. The intensity $|E|^2$ at vacuum/hBN is ~ 12 times larger than that at vacuum/Au. In addition, the maximum intensity of hBN waveguide is not at the surface, but inside hBN, as shown in Fig. RS9a, which is ~ 23 times larger than that at vacuum/Au. It should be noted, for SPP, the ring slit ($m=0$) excited with LCP does not form a focusing point in the center, but a doughnut pattern. To form a focusing point, spiral slit ($m=-1$) should be adopted and we use the focusing point to compare the intensity.

In addition, we can try to measure the loss from the decrease of photoemission intensity during propagation. As shown in Fig. S10a,b, we measured the photoemission intensity along the propagation direction excited from a slit with 410 nm laser at normal incidence. The decay of photoemission intensity along the propagation direction can be observed. Considering the two-photon photoemission process, $P_E \propto I^2 \propto |E|^4$, the electric field $|E|$ along the propagation direction is extracted in Fig. S10c. From $X = 2 \mu\text{m}$ to $X = 7 \mu\text{m}$, the amplitude of $|E|$ (between the red dashed lines) decreases from 0.97 to 0.77, which means the amplitude of $|E|$ has a 20% decrease in $5 \mu\text{m}$. In contrast, in simulations, the amplitude of $|E|$ has a 7.5% decrease in $5 \mu\text{m}$. The larger loss measured in experiments could be due to fabrication quality and the inhomogeneous laser spot. Because the relatively small laser spot ($\sim 150 \mu\text{m}$) was used in normal incidence, it's hard to evaluate the loss accurately, and the measured value should be larger than actual loss."

Comment 7. What are the images in S5a and b. Are the left and right images identical with the right one intensity enhanced?

Reply:

We thank the referee for the helpful comments. Yes, the left and right images are identical in S5a and b. The right images are adjusted with brightness and contrast to show the weak interference fringes around the focusing point.

To make it clear, we revised the caption for original Fig. S5a and b by adding a sentence “The left and right images are identical except that the right images are adjusted with brightness and contrast to show the weak interference fringes around the focusing point”.

Comment 8. In line 189, the author claims to have created the smallest photoemission source, what is its size? In line 190, the authors mentioned that a radius of $r_0=20.7\mu\text{m}$ is adopted here. It is very confusing. The author should clarify that this radius is the size of the waveguide.

Reply:

We thank the referee for the helpful suggestions. We agree with the referee’s suggestions. The size of photoemission source is the size of working area, where electrons are emitted, that is the nanofocusing spot with a size $< 80\text{ nm}$. Why using working area rather than the radius of the whole etched ring? The reason is the working area is the actual region where electrons are emitted, which partly determines the electron coherent features. Therefore, people expect smaller working area in practical applications. The common methods to realize small photoemission sources are using sharp tip photocathodes, such as W, W(ZrO), Ta. The sizes of sharp tip photocathodes are from tens of nanometers to micrometers. As you can see, the size of our designed flat photoemission source is comparable to sharp tip photocathodes. In contrast, the sizes of normal flat photocathodes, such as LaB₆, Au, Cu are tens of micrometers to millimeters. In addition, the common advantage of flat photoemission sources is the smaller photoemission angle, compared with sharp tip photocathodes, which also benefits the coherent features. In summary, our designed photoemission source inherits both advantages of sharp tip and flat photocathodes (small size and small photoemission angle), therefore, is very promising in applications.

To clarify this problem, we revised the corresponding sentences in the main manuscript as “For the lowest-order mode, the spatial distribution of the emitter showed a nanoscale localized spot with strong intensity (Fig. 4a,b), which is the smallest lateral size (< 80 nm) of flat photoemission sources, as reported previously. It should be noted, herein, the size of the photoemission source means the working area, where the electrons are emitted, because the working area partly determines the electron coherent features. In addition, an etched ring with a large radius $r_0 = 20.7$ μm is adopted here, which indicates that the strong focusing spot in the center is powerful evidence for the low-loss properties of the near-field modes.”

Comment 9. In the SI, line 98. Can the authors explain the choice of slit width, are all the simulation done with the slit width of 180nm and how does it affect the simulation?

Reply:

We thank the referee for the helpful comments. The choice of slit width is based on the simulations to realize the optimal coupling efficiency. As shown in Fig. R6, the coupling efficiency is acceptable for slit widths from 150-200 nm. Therefore, we chose 180 nm in our experiments.

Fig. R6. Optimization of slit width by the simulation of the amplitude of $|E|$ coupled into 60 nm hBN waveguide vs slit width. The optimal width are around 180 nm.

To clarify this issue, we add a Fig. R6 in the supplementary material as Figure S4.

REVIEWER COMMENTS

Reviewer #2 (Remarks to the Author):

The authors have largely revised the manuscript and SI according to my previous comments, and nicely addressed some of the concerns. There is still one part regarding photoemission process that is not quite convincing, and most likely incorrect, so far. Therefore, it can not be published in Nat. Comm. or other journals before these are addressed.

Specifically, in the response letter, the authors responded "In addition, considering above discussion and power-dependent measurements, the photoemitted electrons are not from secondary electrons, but attributed to direct two photon photoemission". This is certainly a misunderstanding of the underlying physics of such photoemission process. It is well known that photoemission signals are dominantly from secondary electrons in the low kinetic energy regime, as reported decades ago by Knoesel et al., ref.1 and many others. A simple power dependence would not tell its origin. In fact, if the signal is purely due to two photon photoemission, one would see the 2-photon replica of the valence band, which is also well known.

What's more, the author claimed "while near-vertical photoemission in momentum space, and narrow spread in energy space are simultaneously achieved due to the atomically flat surface of hBN and the two-photon photoemission process". This is a strongly biasing statement, and is incorrect. Such phenomenon is in fact not achieved, but limited by the photoemission. Moreover, according to the band structure of bulk hBN, ref.2 which the authors did not seem to provide even though it is easily accessible in the literature, the dominant photoelectrons should occur along the L-M direction. Therefore, in the reported experiment, only a fraction of the photoelectrons are observed, simply due to the small photoemission horizon set by the limited photon energies. Photoelectrons in other valleys are not observed, at all.

While the proper discussion of the photoemission process will be less novel compared to the claims in the manuscript, it is still recommended that the authors rephrase the text, with proper references added, so that it clearly explains the origin of the observed photoelectron signals in Fig. 4.

1. Knoesel, Ernst, et al. "Dynamics of photoexcited electrons in metals studied with time-resolved two-photon photoemission." *Surface Science* 368.1-3 (1996): 76-81.
2. Blase, Xavier, Angel Rubio, Steven G. Louie, and Marvin L. Cohen. "Quasiparticle band structure of bulk hexagonal boron nitride and related systems." *Physical review B* 51, no. 11 (1995): 6868.

Reviewer #3 (Remarks to the Author):

I would like to thank the authors for carefully revising the manuscript. I am satisfied with the comprehensive response provided by the authors, which has addressed most of my concerns and inquiries.

Based on the experimental findings presented, I am confident that the results are robust and significant, making a great contribution to the scientific community. I recommend publication of this manuscript without any further delay.

We thank the reviewers for their careful reviewing of our manuscript and for their detailed and constructive reports. During the revision, we have taken all the comments from the reviewers into consideration. Our responses to the reviewers' comments are as follows and highlighted in blue.

Report of Reviewer #2

The authors have largely revised the manuscript and SI according to my previous comments, and nicely addressed some of the concerns. There is still one part regarding photoemission process that is not quite convincing, and most likely incorrect, so far. Therefore, it can not be published in Nat. Comm. or other journals before these are addressed.

Specifically, in the response letter, the authors responded “In addition, considering above discussion and power-dependent measurements, the photoemitted electrons are not from secondary electrons, but attributed to direct two photon photoemission”. This is certainly a misunderstanding of the underlying physics of such photoemission process. It is well known that photoemission signals are dominantly from secondary electrons in the low kinetic energy regime, as reported decades ago by Knoesel et al.,ref.1 and many others. A simple power dependence would not tell its origin. In fact, if the signal is purely due to two photon photoemission, one would see the 2-photon replica of the valence band, which is also well known.

What's more, the author claimed “while near-vertical photoemission in momentum space, and narrow spread in energy space are simultaneously achieved due to the atomically flat surface of hBN and the two-photon photoemission process”. This is a strongly biasing statement, and is incorrect. Such phenomenon is in fact not achieved, but limited by the photoemission. Moreover, according to the band structure of bulk hBN,ref.2 which the authors did not seem to provide eventhough it is easily accessible in the literature, the dominant photoelectrons should occur along the L-M direction. Therefore, in the reported experiment, only a fraction of the photoelectrons are observed, simply due to the small photoemission horizon set by the limited photon energies. Photoelectrons in other valleys are not observed, at all.

While the proper discussion of the photoemission process will be less novel compared to the claims in the manuscript, it is still recommended that the authors rephrase the text, with proper references added, so that it clearly explains the origin of the observed photoelectron signals in Fig. 4.

1. Knoesel, Ernst, et al. "Dynamics of photoexcited electrons in metals studied with time-resolved two-photon photoemission." *Surface Science* 368.1-3 (1996): 76-81.
2. Blase, Xavier, Angel Rubio, Steven G. Louie, and Marvin L. Cohen. "Quasiparticle band structure of bulk hexagonal boron nitride and related systems." *Physical review B* 51, no. 11 (1995): 6868.

Reply:

We thank the referee for the very helpful comments and suggestions. We agree with the referee that a simple power dependence would not tell its origin and the secondary electrons should be involved in the photoemission process. We also agree with the referee that photoexcitation should occur in multiple valleys of hBN band structures, including the L-M direction. In addition, interband defect states could also play a role in the photoemission process. Therefore, the secondary electron, intervalley scattering and interband defect states should be discussed in the photoemission process.

We revised the discussions of the photoemission process of hBN in the supplementary materials and added necessary references as follows:

“Phenomenally, the photoemission of hBN with 410 nm laser is a two-photon process, as shown in the power-dependent measurements, i.e., two photons (2×3.02 eV) are absorbed to overcome the work function of hBN. Considering the complicated band structure of hBN, the underlying physics of photoemission should be discussed, including the contributions of secondary electrons, intervalley scattering and interband defect states. It has been reported that photoemission signals are dominantly from secondary electrons in the low kinetic energies for metal, in particular for energies close to Fermi level⁶. For wide-bandgap hBN, secondary electrons could also have a contribution to the low energies of photoemission spectrum, but the contribution could be less important than that in metal due to the large bandgap of hBN. In addition,

photoexcitation should occur in multiple valleys of hBN band structure, including the valleys along L-M direction⁷⁻⁹. In our experiment, only the photoelectrons around the Γ point are observed, simply due to the small photoemission horizon set by the limited photon energies. The intervalley scattering from other valleys toward Γ point could also have a contribution to the final photoemission signals. What's more, defects in hBN could be possibly introduced by sample preparation and laser illumination, the defect states in the wide bandgap could also assist the two-photon process by creating actual intermediate energy levels. Therefore, multiple effects could have contributions to the photoemission process. However, it's hard to evaluate how important of each effect in current stage and is out of the main claims of the manuscript. More detailed investigations on the photoemission mechanism of hBN could be performed in the future.

6. Knoesel, E., Hotzel, A., Hertel, T., Wolf, M. and Ertl, G. Dynamics of photoexcited electrons in metals studied with time-resolved two-photon photoemission. *Surf. Sci.* **368**, 76–81 (1996).
7. Blase, X., Rubio, A., Louie, S.G. and Cohen, M.L. Quasiparticle band structure of bulk hexagonal boron nitride and related systems. *Phys. Rev. B* **51**, 6868 (1995).
8. Hunt, R.J., Monserrat, B., Zólyomi, V. and Drummond, N.D. Diffusion quantum Monte Carlo and G W study of the electronic properties of monolayer and bulk hexagonal boron nitride. *Phys. Rev. B* **101**, 205115 (2020).
9. Artús, L. et al. Ellipsometry Study of Hexagonal Boron Nitride Using Synchrotron Radiation: Transparency Window in the Far-UVC. *Adv. Photonics Res.* **2**, 2000101 (2021).

”

We also revised the sentence in the abstract to make it more accurate. The sentence is revised as “while near-vertical photoemission in momentum space, and narrow spread in energy space are simultaneously observed due to the atomically flat surface of hBN and the small photoemission horizon set by the limited photon energies.”

Report of Reviewer #3

I would like to thank the authors for carefully revising the manuscript. I am satisfied with the comprehensive response provided by the authors, which has addressed most of my concerns and inquiries.

Based on the experimental findings presented, I am confident that the results are robust and significant, making a great contribution to the scientific community. I recommend publication of this manuscript without any further delay.

Reply:

We thank the referee for the recommendation of publication!

REVIEWERS' COMMENTS

Reviewer #2 (Remarks to the Author):

The authors have reasonably revised the manuscript to address my comments regarding the photoemission processes, so that the current manuscript provides a more accurate picture of the observed spectroscopic phenomena. Therefore, I recommend it for publication in Nat. Comm.